# Machine Learning Technology for EEG-Forecast of the Blood–Brain Barrier Leakage and the Activation of the Brain’s Drainage System during Isoflurane Anesthesia

**DOI:** 10.3390/biom13111605

**Published:** 2023-11-02

**Authors:** Oxana Semyachkina-Glushkovskaya, Konstantin Sergeev, Nadezhda Semenova, Andrey Slepnev, Anatoly Karavaev, Alexey Hramkov, Mikhail Prokhorov, Ekaterina Borovkova, Inna Blokhina, Ivan Fedosov, Alexander Shirokov, Alexander Dubrovsky, Andrey Terskov, Maria Manzhaeva, Valeria Krupnova, Alexander Dmitrenko, Daria Zlatogorskaya, Viktoria Adushkina, Arina Evsukova, Matvey Tuzhilkin, Inna Elizarova, Egor Ilyukov, Dmitry Myagkov, Dmitry Tuktarov, Jürgen Kurths

**Affiliations:** 1Department of Biology, Saratov State University, Astrakhanskaya Str. 83, 410012 Saratov, Russia; inna-474@yandex.ru (I.B.); shirokov_a@ibppm.ru (A.S.); terskow.andrey@gmail.com (A.T.); mariamang1412@gmail.com (M.M.); krupnova_0110@mail.ru (V.K.); admitrenko2001@mail.ru (A.D.); eloveda@mail.ru (D.Z.); adushkina.info@mail.ru (V.A.); arina-evsyukova@mail.ru (A.E.); tuzhilkinma@yandex.ru (M.T.); inna.elizarowa7@yandex.ru (I.E.); juergen.kurths@pik-potsdam.de (J.K.); 2Physics Department, Humboldt University, Newtonstrasse 15, 12489 Berlin, Germany; 3Institute of Physics, Saratov State University, Astrakhanskaya Str. 83, 410012 Saratov, Russia; kssergeev@mail.ru (K.S.); nadya.i.semenova@gmail.com (N.S.); slepnevav@sgu.ru (A.S.); karavaevas@gmail.com (A.K.); mdprokhorov@yandex.ru (M.P.); rubanei@mail.ru (E.B.); fedosov_optics@mail.ru (I.F.); paskalkamal@mail.ru (A.D.); egor.re01@mail.ru (E.I.); ivanov.ivao@yandex.ru (D.T.); 4Institute of Radio Engineering and Electronics of RAS, Zelenaya Str. 38, 410019 Saratov, Russia; 5Research Institute of Cardiology, Saratov State Medical University, B. Kazachaya Str. 112, 410012 Saratov, Russia; 6Institute of Biochemistry and Physiology of Plants and Microorganisms, Russian Academy of Sciences, Prospekt Entuziastov 13, 410049 Saratov, Russia; 7Centre for Analysis of Complex Systems, Sechenov First Moscow State Medical University, Bolshaya Pirogovskaya 2, Building 4, 119435 Moscow, Russia; 8Potsdam Institute for Climate Impact Research, Telegrafenberg A31, 14473 Potsdam, Germany

**Keywords:** anesthesia, machine learning technology, spectral power analysis, blood–brain barrier, brain’s drainage system

## Abstract

Anesthesia enables the painless performance of complex surgical procedures. However, the effects of anesthesia on the brain may not be limited only by its duration. Also, anesthetic agents may cause long-lasting changes in the brain. There is growing evidence that anesthesia can disrupt the integrity of the blood–brain barrier (BBB), leading to neuroinflammation and neurotoxicity. However, there are no widely used methods for real-time BBB monitoring during surgery. The development of technologies for an express diagnosis of the opening of the BBB (OBBB) is a challenge for reducing post-surgical/anesthesia consequences. In this study on male rats, we demonstrate a successful application of machine learning technology, such as artificial neural networks (ANNs), to recognize the OBBB induced by isoflurane, which is widely used in surgery. The ANNs were trained on our previously presented data obtained on the sound-induced OBBB with an 85% testing accuracy. Using an optical and nonlinear analysis of the OBBB, we found that 1% isoflurane does not induce any changes in the BBB, while 4% isoflurane caused significant BBB leakage in all tested rats. Both 1% and 4% isoflurane stimulate the brain’s drainage system (BDS) in a dose-related manner. We show that ANNs can recognize the OBBB induced by 4% isoflurane in 57% of rats and BDS activation induced by 1% isoflurane in 81% of rats. These results open new perspectives for the development of clinically significant bedside technologies for EEG-monitoring of OBBB and BDS.

## 1. Introduction

Anesthesia is an integral part of surgery and is one of the greatest achievements of medicine, which allows us to perform complex and long-term surgical procedures painlessly, safely and stably. Anesthesia causes behavioral and electrical changes due to the direct effects of anesthetics on specific receptors in the central nervous system (CNS), leading to a controlled and reversible loss of consciousness and ensuring optimum conditions for patients under surgery [1,2]. However, there is growing evidence that has begun to question the safety of anesthesia, especially in infants and elderly people [1,2,3,4,5,6,7]. Indeed, several clinical and experimental studies have proposed that anesthetic agents induce long-term morphological and functional changes in the brain [8,9,10,11]. There are results that clearly demonstrate a close relationship between the neurotoxicity associated with the development of Alzheimer’s disease and anesthesia [12]. Over recent years, a series of studies have begun to focus on anesthesia-induced neurocognitive dysfunction and neuroinflammation [13,14,15].

At this point, many experimental studies have found that anesthesia can disrupt the integrity of the blood–brain barrier (BBB), leading to neuroinflammation [1,3,16,17,18]. BBB leakage is strongly associated with the development of various chronic neurodegenerative disorders, including Alzheimer’s and Parkinson’s diseases, amyotrophic lateral and multiple sclerosis [19,20]. Postoperative neurocognitive defects, following peripheral surgery under general anesthesia with isoflurane, have received much attention [3,4,21,22,23,24]. Both neuroinflammation and BBB disruption have been observed in subjects after general anesthesia. Furthermore, the anesthetic effects on BBB permeability depend on the duration of anesthesia [25]. Therefore, the control of the effects of anesthesia on BBB integrity during surgery, and of postoperative consequences, has become an important issue of safe medicine. However, there are no reliable methods for the real-time monitoring of the BBB during surgery.

In our previous study, we proposed a new approach for real-time controlling of BBB leakage based on a nonlinear analysis of EEG dynamics and machine learning technology in healthy male rats [26]. We clearly demonstrated that it enables one to find the EEG markers of the opening of the BBB (OBBB) induced by loud sounds. These, our pilot findings, open new perspectives for the development of technologies for real-time monitoring of the OBBB during anesthesia. This also opens a novel approach for forecasting and pre-venting postoperative consequences. Based on our preliminary data, here, we aim to find the EEG markers of the OBBB using 1% and 4% isoflurane anesthesia in in vivo and ex vivo experiments. 

Machine learning methods have often been used to analyze medical and biological data. One such method for data classification is artificial neural networks (ANNs) [27,28,29,30]. This method has proven to be successful for binary and multi-class data classification. The supervised training method assumes that, initially, there is a set of data for which the responses expected from the ANNs are uniquely defined [31]. This dataset is divided into two subsets, most of which are used for training, and the rest (testing dataset) for subsequent testing. During the process of training, ANNs receive examples of input data and the corresponding correct answers. This allows us to determine connections between neurons that give the largest number of correct answers. Then, in the case of correct training, when the network has good accuracy with training and testing data, the trained network can be used to analyze unlabeled data and search for patterns in it that are similar to the training data [31].

For an automatic detection and forecast of the anesthesia-induced OBBB, we used the methods of ANN construction technology and a nonlinear spectrum analysis of EEG recordings obtained from the same anesthetized rats treated with 1% isoflurane and gradually turning into 4% inhalation anesthesia with real-time optical monitoring of the BBB leakage. The ANNs were trained on our previous data of the detection of sound-induced OBBB associated with an activation of the brain’s drainage system (BDS) [26]. A super-vised training of neural networks is often used in medical and biological studies to search for behavioral abnormalities. This means that the dataset is divided into 1: a training set, in which all data are labeled according to the correct recognizing regimes, and 2: a testing set, which the neural network must recognize. The training set is used to train the network with appropriate labels for correct answers. After successful training, the network becomes "smart" and can identify the necessary patterns for the correct answer. In the case of successful learning, the neural network allows one to find out patterns that are not always obvious. This makes neural networks an indispensable tool in data processing.

## 2. Materials and Methods

### 2.1. Subjects

The experiments were conducted on three groups of adult male Wistar rats (250–280 g or 8- to 10-week-old) in the following conditions (I) before isoflurane anesthesia; (II) anesthesia with the general 1% concentration of isoflurane; (III) anesthesia with the lethal 4% dose of isoflurane. All experimental procedures were performed in accordance with the “Guide for the Care and Use of Laboratory Animals” [32], Directive 2010/63/EU on the Protection of Animals Used for Scientific Purposes and the guidelines from the Ministry of Science and High Education of the Russian Federation (No. 742 from 13.11.1984) and have been approved by the Bioethics Commission of Saratov State University (Protocol No. 8, 18.04.2023). The animals were kept in a light/dark environment with the lights on from 8:00 to 20:00 and fed ad libitum with standard rodent food and water. The ambient temperature and humidity were maintained at 24.5 ± 0.5 °C and 40–60%, respectively. The in vivo experiments were performed on the same 53 rats in the following time sequences: 30 min EEG recording before the onset of anesthesia and 30 min EEG recording during the general 1% isoflurane anesthesia, after which the lethal dose of 4% isoflurane anesthesia was applied and the EEG was recorded until the death of the animal (from 1 to 4 h depending on the specific animal). Animals were distributed in such a way that 11 rats were only used for in vivo simultaneous monitoring of OBBB and BDS with the subsequent nonlinear analysis of EEG dynamics and machine learning. Other 42 rats after in vivo monitoring of the BBB permeability to EBDC were used for the additional ex vivo analysis of OBBB using confocal microscopy (n = 21) and spectrofluorometric assay (n = 21) in the following groups: (1) the control, without anesthesia; (2) and (3) the groups treated with 1% and 4% isoflurane anesthesia, respectively. Additionally, 21 rats were used for the study of different doses of isoflurane effects on the BDS activity in the above indicated groups. N = 7 in each group and in all sets of ex vivo experiments.

### 2.2. EEG and EMG Recording

A two-channel cortical EEG/one and EMG (Pinnacle Technology, Taipei City, Taiwan) were used in this study. Ten days before experiments, two silver electrodes were implanted at a depth of 150 µm in coordinates (L: 2.5 mm and D: 2 mm) from Bregma between the skull and the dura matter under anesthesia with 1% isoflurane at 1L/min N_2_O/O_2_—70:30. The EEG leads were fixed with dental acrylic. An EMG lead was implanted in the neck muscle. Ibuprofen (15 mg/kg) was provided in water supply for 3 days after surgery to avoid post-surgery pain. We used self-developed criteria of the EEG patterns and the EEG markers of OBBB, published earlier and presented in Section 2.6 and Section 2.7 [26]. The EEG activity was measured on the same rats before and during anesthesia with the general 1% concentration of isoflurane turning into the lethal 4% dose of this anesthetic agent.

### 2.3. Optical In Vivo and Ex Vivo Analysis of OBBB

In vivo real-time monitoring of extravasation of Evans Blue dye (EBD) from the cerebral vessels into brain tissues was performed using confocal microscopy via the chronic optical windows (ø 3 mm) and the adapted protocol for multiphoton imaging of the cortical cerebral vessels in freely moving rodents [33,34]. A polyethylene catheter (PE-10 tip, Scientific Commodities Inc., Lake Havasu City, AZ, USA) was inserted into the right femoral vein for the EBD injection in a single bolus dose (2 mg∕100 g, 1% solution in physiological 0.9% saline, Sigma Chemical Co., Ltd., St. Louis, MI, USA). Images were captured with Nikon A1R confocal microscope with dry 10 × 0.45 lens (CFI Plan Apo Lambda S 10X, MRD70100 Nikon Corp., Tokyo, Japan) over 60 min with 1 min intervals. At each time point, an image stack of 25 slices within the 120 µm depth range was captured, then maximum intensity Z projection was calculated.

For ex vivo analysis of OBBB, rats were decapitated (in the control group: 30 min after the start of experiment without anesthesia; the group treated with 1% isoflurane: 30 min after onset of anesthesia; in the group treated with 4% lethal dose of isoflurane: 1 or 4 h after the beginning of anesthesia depending on the appearance of in vivo signs of OBBB in each rat). The brains were quickly removed, fixed with 4% neutral buffered formalin for 24 h and cut into 50 µm thick slices on a vibratome (Leica VT 1000S Microsystem, Munich, Germany). Brain sections were processed according to the standard IHC protocol (abcam) with the corresponding primary and secondary antibodies. The slices were incubated for one night with the goat anti-mouse NG2 antibody (1:500; ab 50009, Abcam, Cambridge, UK) and the goat anti-rabbit GFAP antibody (1:500; ab 207165, Abcam, Cambridge, UK). On the next day, after several rinses in phosphate-buffered saline, the slices were incubated for 2 h with fluorescent-labeled secondary antibodies (goat anti-mouse IgG (H + L) Alexa Four 561; goat anti-rabbit IgG (H + L) Alexa Four 488; Invitrogen, Molecular Probes, Eugene, OR, USA). In the final stage, the sections were transferred to a glass and 15 µL of mounting liquid (50% glycerin in PBS with DAPI at a concentration of 2 µg/mL) was applied to the section. The preparation was covered with a cover glass and confocal microscopy was performed. Approximately 8–12 slices per animal from cortical and subcortical regions were imaged.

Brain sections were visualized using a confocal microscope (Nikon A1R MP, Nikon Instruments Inc., Tokyo, Japan) with a ×20 lens (0.75 NA) or a ×100 lens for immersion in oil (0.45 NA). DAPI, Alexa Fluor 488, Alexa Fluor 561 and Evans Blue were excited with excitation wavelengths of 405 nm, 488 nm, 560 nm and 647 nm, respectively. Three-dimensional visualization data were collected by obtaining images in the x, y and z planes. The images were obtained using NIS-Elements software 5.02 (Nikon Instruments Inc.) and analyzed using Fiji software 2.7.0. (Open-source image processing software) and Vaa3D 3.2 (Open-source visualization and analysis software).

### 2.4. Spectrofluorometric Assay of EBD Extravasation

The EBD (2 mg/body weight, 1% solution in physiological 0.9% saline, Sigma Chemical Co., St. Louis, MI, USA) was injected into the femoral vein via polyethylene catheter (PE-10 tip, Scientific Commodities Inc., Lake Havasu City, AZ, USA) and circulated in the blood for 30 min. Then, the rats were decapitated (the time of brain sampling for this experiment was the same as is indicated in Section 2.3) and their brains were quickly collected and placed on ice (no anticoagulation was used during blood collection). Prior to brain removal, the brains were perfused with saline to wash out the remaining dye in the cerebral vessels. The EBD level in brain tissues was evaluated in accordance with the recommended protocol [35] in the control group, without anesthesia, and in the groups treated with 1% and 4% isoflurane.

### 2.5. Optical Monitoring of BDS Activity

To study the BDS activity, we analyzed the fluorescein isothiocyanate–dextran (FITCD) spreading in the cerebral liquor system with its further lymphatic removal from the brain before and after isoflurane anesthesia. Ten days before experiments, the chronic polyethylene catheters (PE-10, 0.28 mm ID × 0.61 mm OD, Scientific Commodities Inc., Lake Havasu City, AZ, USA) were implanted into the femoral vein for the EBD injection (2 mg/body weight, 1% solution in physiological 0.9% saline, Sigma Chemical Co., St. Louis, MO, USA) to fill the cerebral vessels and into the right lateral ventricle (AP, 1.0 mm; ML, – 1.4 mm; DV, 3.5 mm) for the FITCD injection (5 µL, at a rate of 0.1 µL/min, 0.5% solution in saline, Sigma-Aldrich, St. Louis, MI, USA) according to the protocol reported by Devos et al. [36]. The injection of FITCD into the ventricle and EBD in the femoral vein was performed automatically using a syringe injector (QSI, 53311, Stoelting, Wheat Ln, Wood Dale, IL, USA) immediately after the start of experiment without anesthesia (in the control group) or administration of 1% and 4% isoflurane.

The protocol of ex vivo optical study of FITCD spreading in the whole brain and its accumulation in the deep cervical lymph nodes (dcLNs) is described in detail in our publication [26]. Briefly, ex vivo confocal monitoring of FITCD evacuation from the ventricle to dorsal and ventral parts of the brain followed by transport to dcLNs was performed 30 min after its the intra-ventricular injection. Rats were sacrificed, and their brains and dcLNs were carefully removed. The imaging was performed using a confocal microscope (Nikon A1R MP, Nikon Instruments Inc., Tokyo, Japan) with a ×20 objective (0.75 NA). Two lasers (488 nm and 560 nm, respectively) were used for the excitation of FITCD and EBD, respectively, during the confocal imaging. The images were obtained using NIS-Elements software (Nikon Instruments Inc.) and analyzed using Fiji software (Open-source image processing software) [37]. For a quantitative analysis of the intensity signal from FITCD, ImageJ was used for image data processing and analysis.

### 2.6. Artificial Neural Network

One of the methods used in this paper is based on machine learning by means of artificial neural networks (ANNs). As a rule, raw signal data are rarely taken for data analysis using a neural network. Instead, data scientists extract different “features” from analyzed signals [38,39], such as different statistical, entropy, fractal, time-frequency, spatiotemporal and other characteristics. The choice of the correct set of characteristics largely determines the success of ANN application and its final accuracy. For example, in our previous works we show that statistical characteristics and wavelet-transform can be successfully applied to train ANNs for sleep recognition [40] and recognition of OBBB [26,41]. In all these papers, we used calculation of statistical characteristics in a shifting window, and a similar technique will be used in the current paper.

The technique of shifting window consists of the following: each EEG implementation is divided into some number of windows of a current duration. In this paper, the window duration is fixed, δ=60 s. In order to increase the number of training and testing examples and to make the ANN responses smoother, each window is closely shifted with each other to τ=10 s. For example, if there is a five-minute EEG recording (300 s), then the technique of shifting window divides it by 25 windows with implementations, which are further used to calculate the corresponding characteristics.

An important stage is the specification of “features” taken from an analyzed signal. In the present work, we use the method of time-frequency analysis in a shifting window. Using fast Fourier transform (FFT), the power spectrum was calculated for each implementation channel in a shifting window of 60 s. The harmonics of the power spectrum were then used to calculate the energy per bands for particular channels and windows as
(1)$$P_{c,\;\kappa }=\frac{1}{N_{\kappa 2}-N_{\kappa 1}}\sum_{i=N_{\kappa 1}}^{N_{\kappa 2}}F_{i}^{2},$$
where c=1, 2 is an EEG channel number. The index κ=γ, β,α,Θ,δ indicates different frequency bands: γ∈[30 Hz;50 Hz], β∈[12 Hz;30 Hz], α∈[8 Hz;12 Hz], Θ∈[4 Hz;8 Hz],  δ∈[1 Hz;4 Hz]. The values $N_{\kappa 1}$ and $N_{\kappa 2}$ are numbers of the first and last harmonics (sampled from the FFT) of the corresponding frequency band κ; $F_{i}$ is an amplitude of the harmonics of the FFT. The average energy for the five frequency bands γ, β, α, Θ and δ was calculated as a sum of the harmonics divided by their number in a specified frequency band of FFT signal.

As a result, five power values were obtained for each shifting window and for each channel $P_{c,\;\kappa }$. Next, the values for the first and second EEG channels were summed as $X_{\kappa }=P_{1,\kappa }+P_{2,\kappa }$ and then converted by taking their logarithm and normalization to the sum of the energies for all five bands:(2)$${\bar{X}}_{\kappa }=\frac{\ln X_{\kappa }}{\sum_{\kappa }\ln X_{\kappa }}=\frac{\ln X_{\kappa }}{\ln\prod_{\kappa }X_{\kappa }}=\ln_{\prod X}X_{\kappa }.$$

Further, we will call this normalization of the ANN input signal normalization 1. Surprising results were obtained when training ANNs on the training EEG implementations with usual normalization (2) and then applying this ANN to anesthetized rats using an alternative normalization (differs from Equation (2)). When extracting features from the records of rats with anesthesia, the values of $X_{\kappa }$ were normalized not by the sum of harmonics, as in (2), but by the maximum and minimum, as follows:(3)$${\bar{X}}_{\kappa }^{*}=\frac{\ln X_{\kappa }-\ln X_{min}}{\ln X_{max}-\ln X_{min}}=\ln_{\frac{X_{\max}}{X_{\min}}}X_{\kappa }/X_{min}.$$

Further, we will use this normalization, calling it normalization 2, and compare the results for both types of ANN input (2) and (3).

It was a description of how the input signal for ANNs was set. Now, let us move on to the description of the ANN itself. We used a deep fully connected ANN schematically shown in Figure 1. The input layer consists of 5 neurons (according to the number of frequency bands γ, β, α, Θ and δ) and the hidden layer has 150 neurons, while the output layer consists of 2 neurons (according to binary classification). The neurons marked by a purple color in Figure 1 have a bias and an SELU activation function, while the orange output neurons have no bias shift and a softmax activation function.

The number, structure and activation functions of the hidden layer were determined empirically from the point of view of maximizing the accuracy with the least computational complexity. All layers, except for the output one, were subjected to regularization of both connectivity and biases.

ANN training was carried out using the method of back propagation of errors using the libraries Keras and Tensorflow in the Python programming language. When preparing the training and testing datasets, the above features ${\bar{X}}_{\kappa }$ were calculated in a 60 s shifting window with an overlap of 10 s. The window duration of 60 s was chosen empirically as providing the greatest change in accuracy when the window shifted, on the one hand, and a sufficiently large number of training examples, on the other hand.

To train the network, we used the EEG data of rats with normal behavior and the data of rats with artificially opened BBBs with a loud sound. The process of obtaining and analyzing these data was already described in detail in Ref. [26]. It was found that the optimal accuracy of ANN training is achieved when analyzing records from the 1800th second to the 3600th. The data calculated for all rats from the training set (with OBBB and in the normal state) were combined into one dataset, 70% of which were used for training, and the remaining 30% for testing. In the process of ANN training on the EEG records of rats in the normal state and with an artificially opened BBB, this made it possible to obtain a classification accuracy of 85% in testing dataset.

The process of training the ANN is schematically shown in Figure 2. Top panels illustrate the parts of two signals from EEG channels used to assemble the training dataset. As we used the data of Ref. [26], which were marked as “normal behavior” and “artificial OBBB” caused by loud sound [26], the training dataset was divided into these two classes (see bottom left panel in Figure 2, green for normal behavior ‘0’ and purple artificial OBBB ‘1’). With approximately the same amount of data for both classes, we then split all the data for each class into a set used for training (marked with shading in Figure 2) and a set used for testing (marked with dots in Figure 2) in a ratio of 70% to 30%, respectively. Thus, we obtained 1430 examples of normal behavior and artificial OBBB which were further divided into 500 training and 215 testing examples per class. All data were normalized according to normalization 1 (Equation (2)) before sending them to ANN. The process of training an ANN depending on the training epoch is given in the right bottom panel of Figure 2. The final accuracy of binary classification reached 90% on the training dataset and 85% on the testing dataset. We have uploaded our program used for training the network to a GitHub platform [42].

After training and testing, the resulting classifier was applied to EEG implementations of laboratory rats under the influence of isoflurane anesthesia when ANN input signal is normalized according to normalization 1 (Equation (2)) or normalization 2 (Equation (3)). The task was to detect markers of BBB opening in animals under anesthesia.

### 2.7. Spectral Analysis

In this paper, we analyzed the EEG data using spectral analysis methods, focusing primarily on the low-frequency components of the spectrum of EEG signals. Spectrum analysis was carried out in δ, Θ frequency ranges.

For each EEG recording, the time dependence of the average power in each range was plotted. The average power was calculated in 250 s wide sliding windows across the time record with 50 s offset. We used the Welch approach to estimate the spectral power density [43]. Welch’s spectra were calculated in 60 s windows with 30 s offset.

In addition, we provide a comparison of the normalized change in mean power of EEG signals before and after 1% and 4% isoflurane anesthesia. For each animal’s records, we calculated the median power value during the normal state (m_0_) and during normal anesthesia (m_1_). We calculated the normalized change in power in two frequency ranges (Δδ) and (Δθ) using Equation (4). Then, we calculated the sample mean value.
(4)$$\Delta =\frac{m_{1}-m_{0}}{m_{0}}.$$

### 2.8. Statistical Analysis

The results are presented as mean ± standard error of the mean (SEM). Differences from the initial level in the same group were evaluated using the Wilcoxon test. Intergroup differences were evaluated using the Mann–Whitney–Wilcoxon test. The significance levels were set at *p* < 0.05 for all experiments.

## 3. Results

### 3.1. Effects of Different Doses of Isoflurane Anesthesia on BBB Integrity and BDS Functions

In the first step, we studied the effects of general 1% and lethal 4% doses of isoflurane on the EBDC leakage from the cerebral vessels into the brain tissues in the same rats (n = 11) under the EEG control using in vivo confocal microscopy. Since craniotomy can lead to BBB disruption and neuroinflammation [44], we used the technology of optical window [34] for real-time confocal monitoring of the cerebral vasculature through the intact skull (Figure 3a). Our results did not reveal any changes in BBB permeability to EBDC during a 1 h observation of rats undergoing 1% isoflurane anesthesia (Figure 3b). However, increasing the dose of isoflurane to 4% was accompanied by dramatic BBB disruption that was observed 1–2 h before the death of the animals (Figure 3c).

Additionally, we studied the BBB permeability to EBDC in ex vivo experiments using confocal imaging with the markers of neurovascular units and spectrophotofluorometric assay of the EBAC level in the brains of rats treated with 1% and 4% isoflurane. Figure 3d,e clearly demonstrate that 4% but not 1% isoflurane caused OBBB for EBAC. The quantitative analysis revealed a significant increase in the EBAC level in the brain tissues of rats only from the group treated with 4% isoflurane (0.35 ± 0.05 μg/g tissues vs. 0.11 ± 0.01 μg/g tissues, *p* < 0.001 between the group treated with 4% isoflurane and the control without anesthesia; 0.14 ± 0.03 μg/g tissues and 0.11 ± 0.01 μg/g tissues, NS, between the group treated with 1% isoflurane and the control without anesthesia, the Mann–Whitney–Wilcoxon test, n = 7 in each group) (Figure 3f). Figure 3g shows a photo of the brain with a typical white color obtained from rats treated with 1% isoflurane and a blue color of the brain obtained from rats treated with 4% anesthetic agent. This suggests a significant OBBB for EBAC that occurs only in animals receiving the lethal dose of inhalation anesthesia.

There is evidence that isoflurane and other anesthetic agents enhance the drainage of brain tissues that is correlated with specific changes in the EEG dynamics [45]. In our previous study, we found EEG markers of the intensification of BDS during deep sleep and after the OBBB [26]. Taking into account these data, in the second step, we analyzed the effects of different doses of isoflurane on the BDS activity in rats under EEG control (Figure 4). Figure 4a schematically illustrates the design of our experiment. Our results definitely show that isoflurane induced a dose-related increase in the spreading of FITCD from the lateral right ventricle to dorsal and ventral parts of the brain with further accumulation in the dcLNs (Figure 4b–j). Indeed, the quantitative analysis revealed that the intensity of the fluorescent signal from FITCD in the brain and in the dcLNs was significantly higher in anesthetized rats vs. the control animals without anesthesia (between the control without anesthesia and rats treated with 1% isoflurane: 0.18 ± 0.01 a.u. vs. and 0.11 ± 0.01 a.u., *p* < 0.001 in the dorsal part of the brain; 0.22 ± 0.03 a.u. vs. and 0.12 ± 0.02 a.u., *p* < 0.001 in the ventral part of the brain; 0.26 ± 0.04 a.u. vs. and 0.12 ± 0.02 a.u., *p* < 0.001 in the dcLNs; between the control without anesthesia and rats treated with 4% isoflurane: 0.61 ± 0.03 a.u. vs. and 0.11 ± 0.01 a.u., *p* < 0.001 in the dorsal part of the brain; 0.63 ± 0.04 a.u. vs. and 0.12 ± 0.02 a.u., *p* < 0.001 in the ventral part of the brain; 0.61 ± 0.05 a.u. vs. and 0.12 ± 0.02 a.u., *p* < 0.001 in the dcLNs. Mann–Whitney–Wilcoxon test, n = 7 in each group). However, the effects of isoflurane on the BDS was significantly higher in rats treated with 4% isoflurane than in animals treated 1% with isoflurane (0.61 ± 0.03 a.u vs. 0.18 ± 0.01 a.u., *p* < 0.001 in the dorsal part of the brain; 0.63 ± 0.04 a.u. vs. 0.22 ± 0.03 a.u. in the ventral part of the brain; 0.61 ± 0.05 a.u. vs. 0.26 ± 0.04 a.u. in the dcLNs. Mann–Whitney–Wilcoxon test, n = 7 in each group).

Thus, both series of experiments clearly demonstrate differences in the effects of general and lethal dose of isoflurane on BBB integrity and BDS activity. The general concentration of isoflurane (1%) does not affect the BBB but enhances the BDS. The lethal dose of anesthetic drug causes dramatic OBBB before the death of animals that is accompanied by a significant increase in the BDS functions. These functional models of isoflurane-induced modulation of BBB permeability and BDS activity were used for further spectral analysis of the EEG dynamics and ANNs with the aim of extracting the EEG markers of the OBBB.

### 3.2. Analysis of OBBB with ANN

After training an ANN on the same data as described in detail in Section 2.6 of the Materials and Methods and published in Ref. [26], we applied this network to recognize the OBBB in rats with 1% and 4% isoflurane anesthesia. For training the ANN, we used the functional models of transversal OBBB by sound as well as the natural and induced activation of the BDS during sleep and after the OBBB, respectively [26]. Thus, our ANN is trained to recognize both the OBBB and changes in BDS activity. This will be important for further discussion of the results since our trained network responds to both effects.

When applying our ANN with input normalized according to Equation (2) (normalization 1), we found an interesting result that the ANN recognizes the changes in the brain induced by the general 1% concentration of isoflurane associated with BDS activation but not OBBB (Figure 5a,b). This allows us to assume that this type of ANN with normalization 1 is likely to be more sensitive to the EEG-related changes in response to the activation of movement of brain fluids. Figure 5a,b illustrate the implementations of ANN responses in real time for the two most representative rat implementations under 1% isoflurane anesthesia. Here, we show three main stages: 1—before anesthesia (white background), 2—the general concentration of 1% isoflurane (green background) and 3—the lethal 4% dose of isoflurane (pink background). Orange lines indicate the pure ANN output recorded from the output neuron marked by symbol * in Figure 1.

As can be seen from the first white region of Figure 5a,b, there are too many outliers, making the answer not entirely accurate and incomprehensible. In order to obtain a clearer answer, we added the averaging of this response and obtained black lines. Averaging was carried out in a sliding window lasting 16 min, shifting along the time series with a step of 10 s. From these data, one can obtain a simple binary answer (1—BDS is activated, 0—not) in the next way. If the black curve goes higher than the average level (dash-dotted line) then the BBB is opened or the BDS is activated, while if it falls below the average level, this indicates the opposite: IBBB and an inactive BDS. The above-described binary answer is shown by blue dashed curves in Figure 5a,b.

Figure 5a,b contain only the two most representative animals, but for the other nine rats we found the same ANN response to 1% isoflurane anesthesia. This suggests that the trained ANN with normalization 1 (Equation (2)) recognizes BDS activation well. Therefore, for all animals, we are able to recognize BDS activation during 1% isoflurane anesthesia using the suggested ANN with normalization 1, while this algorithm does not work well for the lethal 4% dose of isoflurane (Figure 5b, for example). Figure 6 illustrates a purple statistical box plot corresponding to ANN responses for all animals. The purple box plot located in a pink area with 4% isoflurane anesthesia is higher than that in the normal condition before anesthesia, but lower than the general 1% isoflurane treatment. This is in full accordance with our assumption that using the ANN with normalization 1 we can see only the BDS activation but not the OBBB.

From the results presented in Section 3.1, the OBBB was only observed in rats under 4% isoflurane. In order to understand this effect, we suggest a second normalization of ANNs based on Equation (3). This approach does not allow one to recognize the BDS activation and, therefore, such a network would not respond to 1% isoflurane associated with BDS activation and the unchanging BBB integrity. The corresponding time implementations for the two most representative rats are given in Figure 5c,d. They are performed in the same manner as in the panels Figure 5a,b but with a red dashed binary answer since this ANN is responsible for recognizing slightly different modes. As can be seen from Figure 5c,d, the suggested ANN responds exclusively to 4% isoflurane. These results coincide with those presented in Section 3.1 and indicating the OBBB in rats under 4% anesthesia. This is confirmed by the statistics for all animals illustrated by the orange box plots in Figure 6. The box plots prepared for the stages before anesthesia and under 1% isoflurane are similar to each other and lie very close to zero, while the box plot for 4% isoflurane is clearly higher. This allows us to conclude that the proposed ANN with normalization 2 allows us to recognize the OBBB. Thus, the ANN responds well to the lethal dose of 4% isoflurane anesthesia associated with a dramatic OBBB.

### 3.3. Power Spectral Analysis of OBBB

We observed qualitatively similar changes in the mean spectral power density in all animals. The administration of general 1% isoflurane anesthesia caused an increase in the mean spectral power density in both the δ and θ frequency ranges. The lethal 4% dose of isoflurane caused the spectral power density to diminish to zero. Typical dynamics of the spectral power density in the δ and θ frequency ranges under the effects of different doses of anesthesia are shown in Figure 7.

On average, under 1% isoflurane anesthesia, the EEG spectral power density in the δ range increases (Figure 8a). However, the changes in the θ frequency ranges observed in animals after the administration of 1% isoflurane are less pronounced compared with the changes in the δ frequency range (Figure 8b).

The Δδ and Δθ indexes show the growth in the EEG spectral power density in the Δδ index after the administration of general 1% isoflurane anesthesia, while the changes in the Δθ index were less pronounced (Figure 9).

## 4. Discussion

In this pilot study on male rats, we analyzed different doses of isoflurane anesthesia, widely used in surgery, on BBB integrity and BDS activity by using the power spectral analysis and ANNs. The ANNs were trained on our previous models of sound-induced OBBB and BDS activation during sleep. Our findings clearly demonstrate that the general 1% isoflurane anesthesia is accompanied by an increase in the FITCD spreading in the brain and its accumulation in the dcLNs vs. a control without anesthesia. There were no changes in the BBB permeability in all rats treated with 1% isoflurane. However, an increase in the dose of anesthesia to 4% is associated with a dramatic BBB leakage and a higher FITCD distribution in the liquor system of the brain and in the dcLNs. Thus, our data revealed the dose-related effects of isoflurane on the BBB and BDS. The general 1% concentration of isoflurane only causes BDS activation, while the high 4% dose of anesthesia induces significant BBB disruption associated with an intense extravasation of EBDC from the cerebral vessels into the brain, causing a higher intensification in the BDS. In our previous study, we demonstrated BDS activation immediately after the OBBB, leading to the removal of molecules that crossed the OBBB from the brain into the peripheral lymphatics [26,45,46]. This can explain the higher activation of the BDS in rats under 4% isoflurane inducing the OBBB compared with a 1% anesthesia without the OBBB. Thus, both doses of isoflurane, general and lethal, are characterized by the BDS activation that was evaluated as the FITCD spreading in the cerebral liquor system and its lymphatic removal from the brain. 

There are conflicting data on the effects of isoflurane on the glymphatic system, which is obviously due to different technical approaches, anesthesia doses and exposure times, and tracers used [47,48,49]. Note that the glymphatic system is considered a route for removing metabolites and toxins from brain tissue via the aquaporin channels, while the underlying mechanisms remain unknown and debated [50,51,52,53,54]. In our study, we aimed to study the effects of isoflurane exclusively on the BDS and OBBB using the classical idea of preferred routes for the movement of brain fluids along the perivascular spaces (PVSs). We assume that the size of the PVSs increases after the OBBB for the water, causing BDS activation [45,46,50,51,55,56]. It is believed that an increase in the volume of the PVSs contributes via the drainage of fluids and the removal of water-soluble metabolites from the brain [57].

Using a power spectral analysis, we found that BDS activation under the general 1% isoflurane anesthesia is accompanied by specific changes in the low-frequency EEG activity (0.1–0.4 Hz). These data are consistent with our previous results showing EEG markers of high BDS activity in the delta spectral power [26].

A machine learning method such as ANNs has proven to be the most effective for the study of the isoflurane effects on the BDS and OBBB. The ANN was trained to recognize both the OBBB and BDS activity based on our functional models of the OBBB and BDS activation in different physiological conditions [26]. Using original approaches, our results show that the first normalization-adapted ANN slightly recognizes the BDS activation under the general 1% isoflurane anesthesia without the OBBB, while the second normalization-configured ANN identifies a dramatic BBB disruption associated with death of rats treated with a lethal dose of anesthetic agent.

The mechanisms of changes of EEG dynamics associated with the OBBB and BDS are not fully understood [26]. The astrocytes play a crucial role in the drainage of brain tis-sues [58]. Indeed, the astrocytes rapidly and significantly change their volume, making a decisive contribution to the change in the total proportion of size of PVSs and the volume of brain fluids in PVSs [59,60,61,62]. Recently, astrocyte activity has been found to be reflected in the low-frequency band < 1 Hz [63,64] that is associated with BDS activation as found in this study and in our previous research [26]. So, Ikeda et al. reported that the low-frequency band below 1 Hz reflects astrocyte-derived depolarization, while high-frequency oscillations above 200 Hz reflect neuron activity [64]. Kurodo et al., using a planar microelectrode array, revealed slow waveforms of spontaneous human astrocyte activity in direct current potentials < 1 Hz [63]. Thus, the spontaneous electrical activity of astrocytes can specifically affect the EEG dynamics, creating an EEG profile in the BDS. The astrocytes are also essential for the formation and maintenance of the BBB [65,66]. It is assumed that the OBBB can affect the EEG behavior via the astrocytes [26,67]. The astrocytic mechanism of EEG modulation can be mediated via astrocyte-mediated regulation of the synaptic conductance [68,69,70], which is involved in electrically induced EEG-activated states in cortical neurons [71].

## 5. Conclusions

In this study, we used ANNs trained on our previous data of sound-induced OBBB [26] and applied them to analyze the EEG data obtained during the general 1% isoflurane anesthesia associated with BDS activation but without the OBBB and the lethal 4% dose of anesthetic agent characterized by a dramatic OBBB and a strong intensification in the BDS. We developed two algorithms to recognize different stages of anesthesia. The first one, based on normalization 1 of Equation (2), allowed us to recognize BDS activation in rats treated with the general 1% concentration of isoflurane. The same ANNs but with normalization 2 of Equation (3) successfully recognized the OBBB in rats under the lethal 4% dose of isoflurane. Based on our hypothesis, we can see that ANNs with normalization 1 recognize BDS activation in 81% of rats treated with 1% anesthesia, and in 57% of rats treated with 4% anesthesia. Similar ANNs but with normalization 2 help to recognize 18% of the OBBB in data with 4% anesthesia, while this level tends to zero for 1% anesthesia. Data obtained from ANNs were confirmed using in vivo and ex vivo optical monitoring of the OBBB and BDS as well as with a nonlinear analysis of EEG dynamics. The results of this interdisciplinary study open new prospects for the development of significant clinical bedside technologies for EEG-monitoring of OBBB.

## 6. Suggestions for Future Studies

The real-time monitoring of the BBB permeability using EEG markers has a great promise for clinical practice, especially for assessing the effects of anesthesia on the BBB during surgical procedures. To verify these experimental studies, it is necessary to conduct clinical investigations to search for the EEG markers of BBB permeability under different types of anesthesia, as well as to evaluate the effectiveness of the methods for the opening of the BBB. For example, EEG markers of the BBB may be useful for assessing the effects of mannitol on the BBB, which is used in patients with glioblastoma to deliver anticancer drugs [72]. The EEG markers of the OBBB will be useful in assessing the effectiveness of the opening of the BBB using focal ultrasound, for example, in patients with Alzheimer’s disease [73]. Since newborn children and the elderly are at risk of developing negative consequences caused by the opening of the BBB after prolonged anesthesia during a surgical procedure [1,2,3,4,5,6,7], it is important to study age-related features in the EEG markers of the OBBB. Because EEG is non-invasive and can be easily applied at the patient’s bedside, EEG markers of the OBBB can be used to assess brain recovery and the effectiveness of a therapy in the post-surgical or post-traumatic periods.

Of particular interest is the physiological explanation for changes in the energy of frequency bands under different states of the BBB depending on external influences. From the point of view of building a classifier, there are two possible directions for improvement. On the one hand, this is the optimization of ANNs to find a relationship between the input data and the network response in order to more accurately and quickly determine the state of the BBB. On the other hand, it is possible to use other classifiers to solve the problem considered in the article, e.g., k-means, KNN, SVM, etc.

## Figures and Tables

**Figure 1 biomolecules-13-01605-f001:**
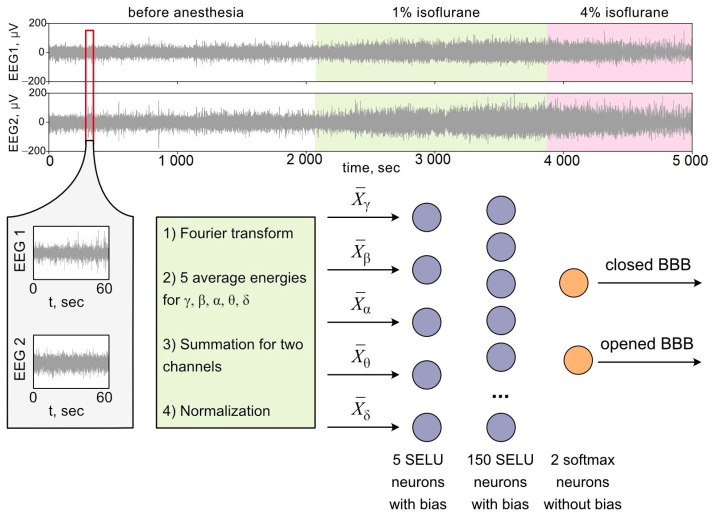
The scheme of experimental data processing and ANN structure.

**Figure 2 biomolecules-13-01605-f002:**
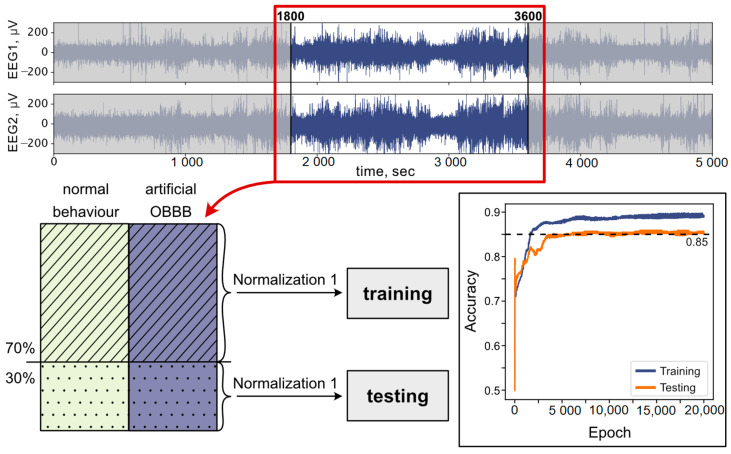
The scheme of ANN training with two dependencies of accuracy on training epoch for training and testing datasets.

**Figure 3 biomolecules-13-01605-f003:**
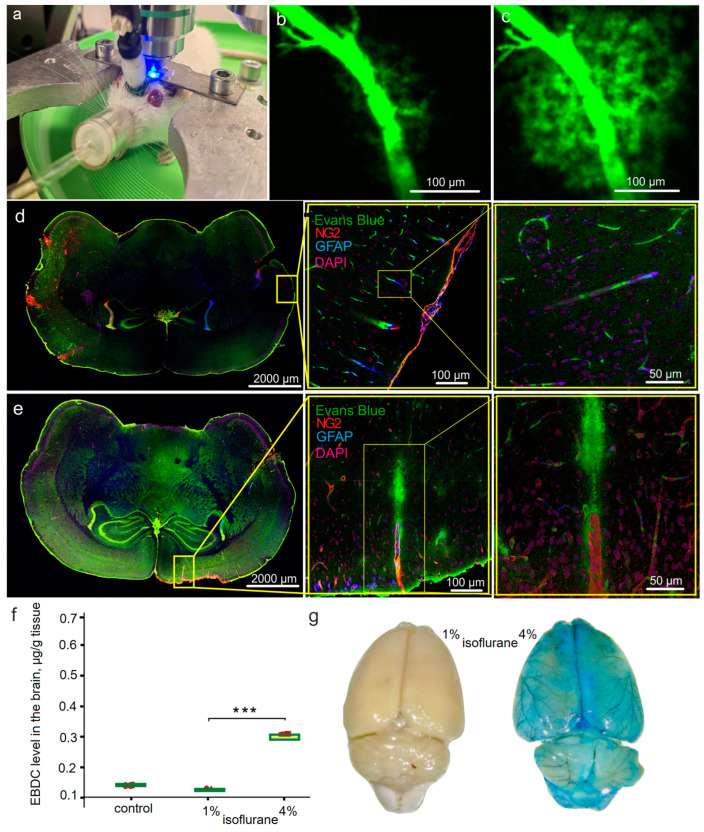
The effects of different doses of isoflurane on BBB permeability to EBDC: (**a**) photo of a real-time confocal microscopy of BBB integrity under EEG control in anesthetized rat; (**b**,**c**) representative images of the cerebral vessels with intact (IBBB) in rats treated with 1% isoflurane and OBBB in rats treated with 4% isoflurane; (**d**,**e**) representative images of whole-brain slices with high resolution of the region of interest illustrating IBBB (**d**) and OBBB for EBAC (**e**); (**f**) quantitative analysis of the EBAC level in the brain of rats from the control group without anesthesia and from the groups treated with 1% and 4% isoflurane, the Mann–Whitney–Wilcoxon test, n = 7 in each group, ***—*p* < 0.001; (**g**) photos illustrating typical white color of the brain of rats treated with 1% isoflurane (IBBB) and blue color of the brain of rats treated with 4% isoflurane (significant OBBB for EBAC).

**Figure 4 biomolecules-13-01605-f004:**
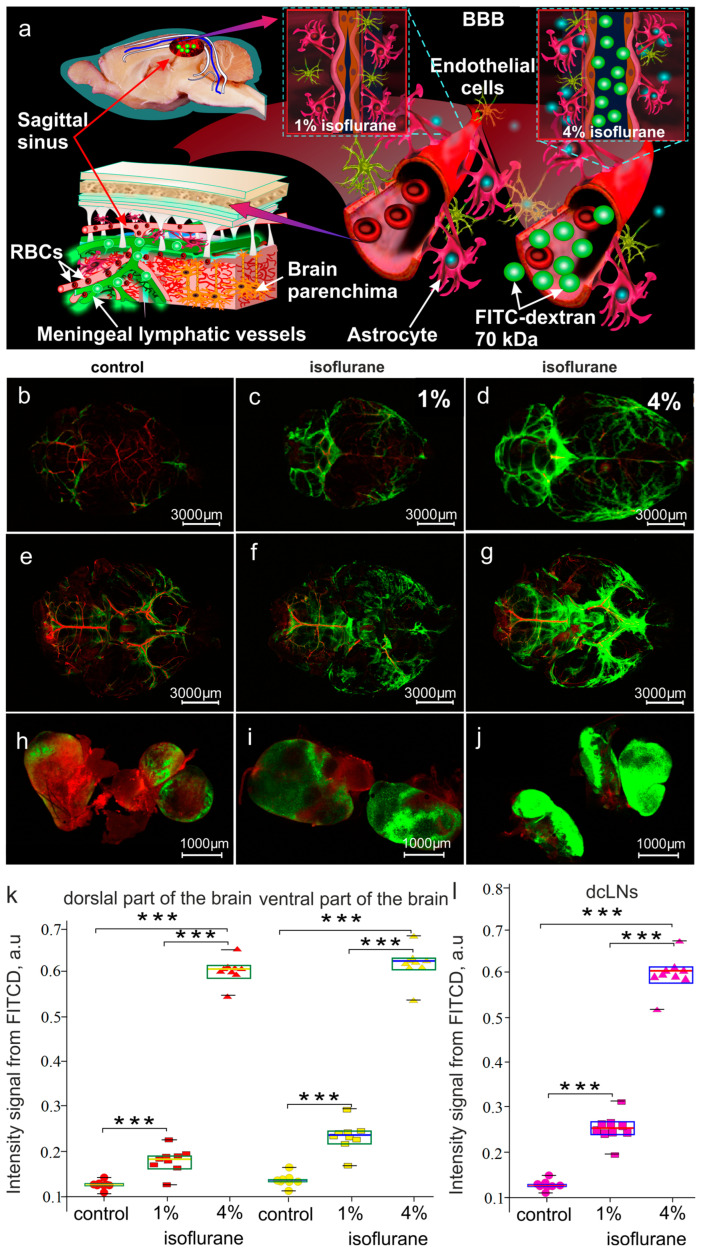
The effects of different doses of isoflurane on BDS activity: (**a**) schematic illustration of design of experiments; (**b**–**j**) representative images of FITCD distribution in dorsal (**b**–**d**) and ventral (**e**–**g**) parts of the brain as well as in dcLNs (**h**–**j**) in the control group (**b**,**e**,**h**) and in the group treated with 1% (**c**,**f**,**i**) and 4% (**d**,**g**,**j**) isoflurane; (**k**,**l**) quantitative analysis of intensity of fluorescent signal from FITCD in dorsal and ventral parts of the brain as well as in dcLNs in the tested groups, respectively; Mann–Whitney–Wilcoxon test, n = 7 in each group, ***—*p* < 0.001.

**Figure 5 biomolecules-13-01605-f005:**
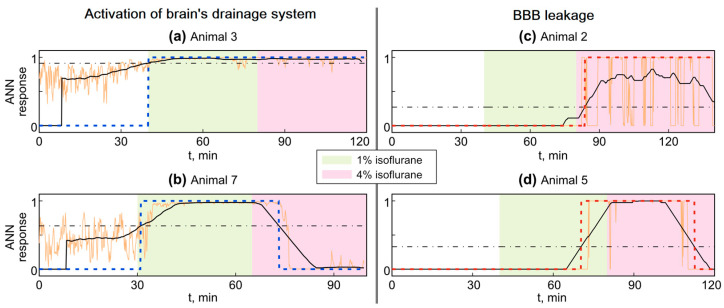
Comparison of temporal implementations of ANN responses (Figure 1) for anesthetized animals when ANN input is normalized according to Equation (2) (**a**,**b**) and when it is normalized according to Equation (3) (**c**,**d**). White, green and pink backgrounds indicate three different conditions, including before anesthesia and under general 1% and lethal 4% doses of isoflurane. Orange lines correspond to the direct response of the ANN, black lines are the average response to remove outliers, blue and red dashed lines represent the binary response with a threshold (it is equal to the average, indicated by a dash-dotted line). In blue curve 1, the BDS is activated; in 0, it is not. In the red curve, 1—OBBB, 0—IBBB or death.

**Figure 6 biomolecules-13-01605-f006:**
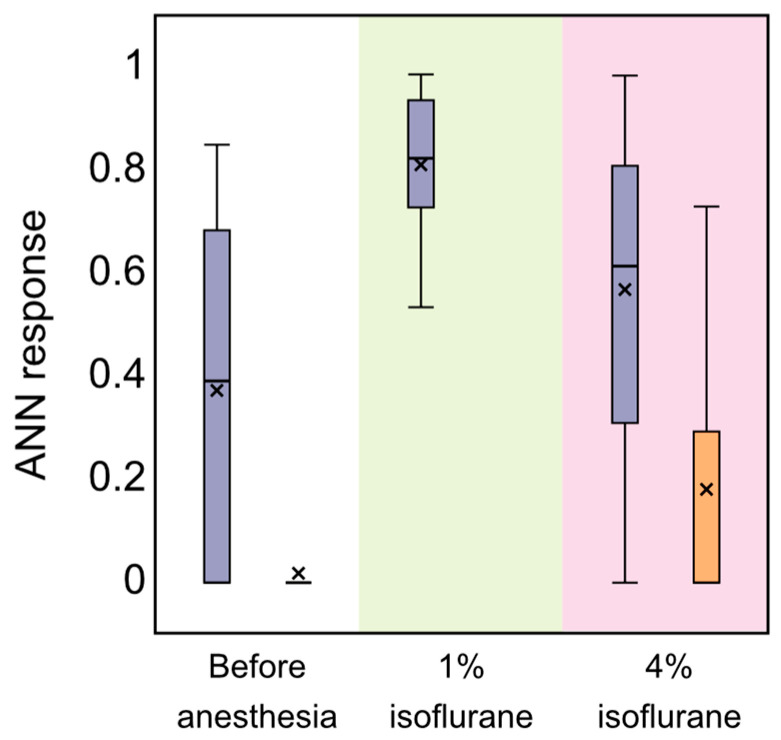
The statistics of nonzero ANN responses to BDS activation (purple box plots) and the OBBB (orange box plots). In this figure, the background color indicates the conditions, including before anesthesia (white) and under general 1% isoflurane anesthesia (green) and lethal 4% concentration of isoflurane administration (pink). In order to calculate the statistics, we used two normalizations of input signals based on Equation (2) (left purple box plots) and on Equation (3) (right orange box plots) for all animals with isoflurane anesthesia.

**Figure 7 biomolecules-13-01605-f007:**
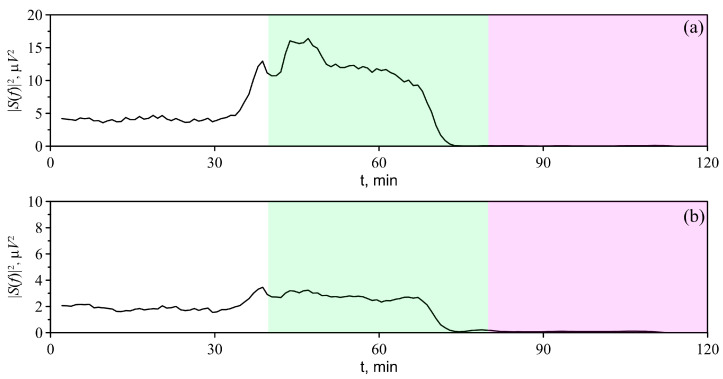
Dynamics of the average EEG power (in animal No. 5 under anesthesia), estimated in time-domain sliding windows in frequency ranges (**a**)—δ, (**b**)—θ. White, green and pink backgrounds indicate the time before anesthesia, 1% (general) and 4% (lethal) isoflurane anesthesia, respectively.

**Figure 8 biomolecules-13-01605-f008:**
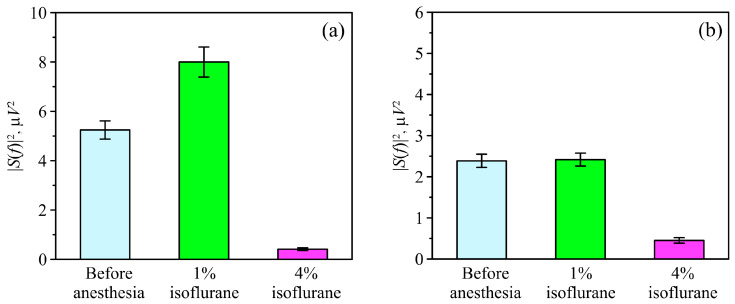
The average values of the spectral power density in the δ (panel (**a**)) and θ (panel (**b**)) frequency ranges of the animal EEG recordings, which were measured under the effects of different doses of isoflurane anesthesia. The height of the bars shows the sample mean power.

**Figure 9 biomolecules-13-01605-f009:**
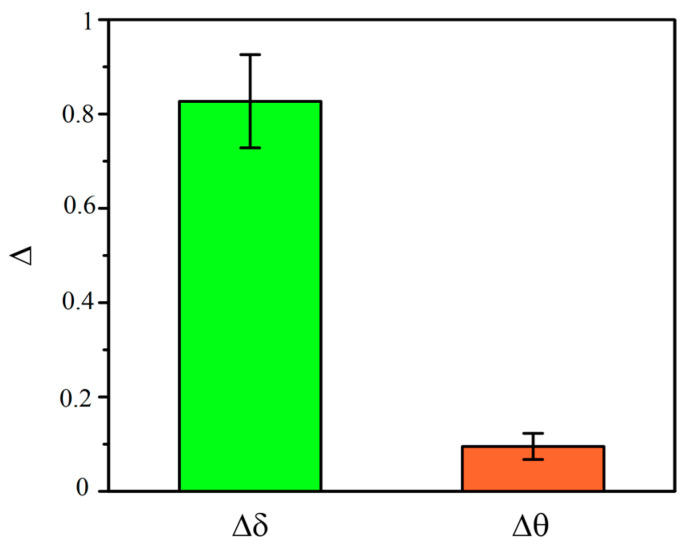
Changes in the normalized power spectral density induced by the administration of general 1% isoflurane anesthesia in the δ and θ frequency ranges. The height of the bars shows the sample mean change in power.

## Data Availability

The data that support the findings of this study are available on request from the corresponding author.

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
