# Peer review of "Machine Learning Technology for EEG-Forecast of the Blood–Brain Barrier Leakage and the Activation of the Brain’s Drainage System during Isoflurane Anesthesia"

_biomolecules, 2023, doi:10.3390/biom13111605_

Round 1
Reviewer 1 Report
Dear authors of the manuscript titled "Machine learning technology for EEG-prognosis of opening of the blood-brain barrier and the activation of brain's drainage system during isoflurane anesthesia,"
After reviewing your manuscript, we are pleased to inform you that it was highly interesting and impressive. The study is well-written, well-organized, and provides thematic illumination.
The abstract is well-crafted, and the content and findings are presented with care. However, minor revisions are needed before publication. These revisions are as follows:
-
The abstract should concisely summarize all the findings, including performance results and key performance metrics.
-
As the article focuses on artificial intelligence and artificial neural networks, it is recommended to provide a brief explanation of this topic to familiarize the audience with artificial neural networks. To achieve this, we suggest referencing the following articles:
- "A hybrid particle swarm and neural network approach for detection of prostate cancer from benign hyperplasia of the prostate."
- "Clinical decision support system for early detection of prostate cancer from benign hyperplasia of the prostate."
- "Efficient Model for Coronary Artery Disease Diagnosis: A Comparative Study of Several Machine Learning Algorithms."
Including these references will enhance the richness of citations in your manuscript.
-
Additionally, since this manuscript presents novel findings, it is important to include a section in the conclusion titled "Suggestions for future studies." This section will provide a roadmap for future research in related areas.
In conclusion, with minimal revisions, this manuscript can be published. We kindly request that you apply these revisions attentively.
Author Response
The authors would like to express their sincere gratitude to the reviewer for the constructive advice, which significantly improved the quality of our article. All changes in the article are highlighted in yellow.
Comment: The abstract should concisely summarize all the findings, including performance results and key performance metrics.
Response: We agree with this remark. We have added more important information to the abstract (Lines 31-49).
Comment: As the article focuses on artificial intelligence and artificial neural networks, it is recommended to provide a brief explanation of this topic to familiarize the audience with artificial neural networks. To achieve this, we suggest referencing the following articles…
Response: We agree with this reviewer's comment. It was necessary to provide more introductory information about the applied method of constructing an ANN. We have added one penultimate paragraph to the introduction with a corresponding description of the ANNs (Lines 92-103).
Comment: Additionally, since this manuscript presents novel findings, it is important to include a section in the conclusion titled "Suggestions for future studies." This section will provide a roadmap for future research in related areas.
Response: We are very grateful to the referee for this remark. We have added the required subsection to the end of conclusion section (Lines 589-610).

Reviewer 2 Report
This paper attempts to build a machine learning model that uses EEG data as input to predict opening of the blood-brain barrier (OBBB). Notably, the authors mention that the method of normalization was critical in their study. The content is strong and warrants publication as it is likely to be of interest to readers. However, there are several areas for improvement:
The method of presentation could be improved. In particular, it should be clearer which data were used for training and which were used for testing. An overview, preferably in the form of a diagram, should be provided to clarify the training and testing processes of the machine learning model.
Even though it's a pilot study, it's necessary to show the accuracy of the model. In particular, numerical data should be provided to illustrate the extent to which the prediction accuracy varies between the two normalization methods.
I would like the authors to publish their script on platforms like GitHub as much as possible. This is necessary for the community to verify the validity of the method used in the study.
Author Response
The authors thank the reviewer for the useful recommendations and advice, which helped us improve the quality of our article for its possible publication in Biomolecules. All changes in the article are highlighted in yellow.
Comment: The method of presentation could be improved. In particular, it should be clearer which data were used for training and which were used for testing. An overview, preferably in the form of a diagram, should be provided to clarify the training and testing processes of the machine learning model.
Response: We are very grateful to the referee for this remark. We have added a schematic illustration of ANN training in Fig. 2 and a description of this process (Lines 305-322).
Comment: Even though it's a pilot study, it's necessary to show the accuracy of the model. In particular, numerical data should be provided to illustrate the extent to which the prediction accuracy varies between the two normalization methods.
Response: Our the accuracies of our ANN were about 90% on training dataset and 85% on cross-validation dataset. It is illustrated in Fig. 2 and additionally mentioned in the text of paragraph before Fig. 2. As for the accuracy on data with anesthesia, we cannot talk about accuracy here, since we do not have data that can be considered a standard. However, based on the performance of our network, we can see that the ANN with normalization 1 recognizes activation of the lymphatic drainage system in 81% of data samples with 1% anesthesia, and in 57% of data samples with 4% anesthesia. Similar ANN but with normalization 2 helps to recognize 18% of OBBB in data with 4% anesthesia. This level tends to zero for 1% anesthesia. We have added relevant information in the abstract and in the conclusion (Lines 45-48 and 580-584).
Comment: I would like the authors to publish their script on platforms like GitHub as much as possible. This is necessary for the community to verify the validity of the method used in the study.
Response: We have uploaded our program to the GitHub platform. This is noted in the paragraph before Fig. 2 and available at the link in Ref. 42 (Lines 318-319).

Round 2
Reviewer 2 Report
Thank you for your revision. Your manuscript is significantly improved.However we need one minor revision. Your presented method in Figure2 is not Cross validation but simple train-test split.Cross-validation is a statistical technique used in machine learning to assess how well a predictive model will generalize to unseen data. Here is a concise explanation of cross-validation:
Objective: Cross-validation helps in estimating the model's effectiveness, tuning hyperparameters, and mitigating overfitting.
Process:
1.The dataset is divided into ‘k’ subsets.
2.The model is trained ‘k’ times, each time using ‘k-1’ subsets for training and the remaining subset for testing.
3.This process helps in ensuring that every data point is used for validation exactly once.
We would prefer to do cross-validation for your study (The average Accuracy of K-fold should be presented.).
Author Response
The authors thank the reviewer for his quick response and comments, with which we agree. Our study was built on our previous results published here (Computational and Structural Biotechnology Journal 21 (2023) 758–768 https://doi.org/10.1016/j.csbj.2022.12.019). In the present manuscript, we have improved our ANN resulting in enlarged training and testing accuracies during detecting sound-induced BBB opening in rats. But in the current work, we are mainly focused on the applied aspect using the same strategies, to solve the problem of searching for an open BBB induced by anesthesia. Despite the fact that the proposed cross-validation can be very useful for our study, we will be happy to do it in our future research but not in this stage because this requires retraining the network and redoing almost all the figures. But, we are required to re-submit our paper within 5 days. Moreover, based on the small difference between our training and testing accuracies, we believe that using the proposed cross-validation technique would not improve our classifier significantly. We improved the text of manuscript and replaced «cross-validation» to «testing» (highlighted in blue) as well as we improved Fig. 2. The authors once again express their gratitude for the opportunity to improve the quality of our article with the help of the reviewer's important advices for its possible publication in Biomolecules.